# Evaluation of the Body Composition and Selected Physiological Variables of the Skin Surface Depending on Technical and Tactical Skills of Kickboxing Athletes in K1 Style

**DOI:** 10.3390/ijerph182111625

**Published:** 2021-11-05

**Authors:** Łukasz Rydzik, Tadeusz Ambroży, Zbigniew Obmiński, Wiesław Błach, Ibrahim Ouergui

**Affiliations:** 1Institute of Sports Sciences, University of Physical Education, 31-571 Krakow, Poland; tadek@ambrozy.pl; 2Department of Endocrinology, Institute of Sport-National Research Institute, 01-982 Warsaw, Poland; zbigniew.obminski@insp.pl; 3Faculty of Physical Education & Sport, University School of Physical Education, 51-612 Wroclaw, Poland; wieslaw.blach@awf.wroc.pl; 4High Institute of Sport and Physical Education of Kef, University of Jendouba, Jendouba 8189, Tunisia; ouergui.brahim@yahoo.fr

**Keywords:** skin temperature, skin pH, kickboxing contest, body composition

## Abstract

Background: Kickboxing is a combat sport with high demands on fitness and coordination skills. Scientific research shows that kickboxing fights induce substantial physiological stress. Therefore, it is important to determine the body composition of athletes before competitions and to analyze the skin temperature and skin pH during the fight. Methods: This study aimed to determine the body composition, skin temperature, and skin pH in kickboxers during a fight according to K1 rules. A total of 24 kickboxers (age range: 19 to 28 years) competing in a local K1 kickboxing league participated in the present study. Results: Changes in skin temperature and pH were observed and significant correlations were found between body composition and weight category. Conclusions: Changes in skin temperature and pH were demonstrated after each round of the bout. Level of body fat and muscle tissue significantly correlates with technical-tactical skills of the K1 athletes studied.

## 1. Introduction

Kickboxing is a physically demanding combat sport with a high focus on coordination skills, with many systems and organs of the human body involved during the fight [1]. To score points and defeat the opponent, kickboxers use both upper limbs, in an attempt to deliver punches, and lower limbs, used to perform kicks [2]. The K1 formula places the least restrictions on the techniques used and the strength of their execution. It allows for all kickboxing techniques to be performed without any restrictions on the strength of the blows. The competitors fight with naked torsos in shorts, wearing helmets on their heads, gloves on hands, and shin and feet guards. Fights, according to the regulations of the World Association of Kickboxing Federation (WAKO), last 3 × 2 min. A bout can be won by knockout or point advantage [3]. Kickboxers are, therefore, required to have an optimal motor and technical skills level and be able to perform specific tactical actions during the fight [4]. However, an important element in sports competition is the body mass and body composition of the athletes and the correct ratios of anthropometric characteristics [5]. These aspects do not only determine the competition with opponents of a similar level of somatic predispositions (i.e., division into weight classes), but also ensure an optimal basis for mastering and using fighting techniques during competition [6]. The adequate level of body composition (i.e., low body fat with the body mass close to the upper threshold for the given class) and the right length ratios may indicate the athlete’s aptitudes and determine his or her sports success [6,7]. In the world of modern sport, body composition monitoring is a basic activity that allows for the evaluation of body changes resulting from practicing a particular sport and determining its participation as the basis for sports selection [8]. The necessity to compete in a sport with weight classes such as K1 rules also requires kickboxers to regularly control their weight and body composition. Based on the diagnosed parameters of body composition, it is possible to precisely determine training (e.g., to develop the range of techniques used and the tactical way of fighting) and a nutritional plan (i.e., to optimize body mass and composition) for kickboxers [8]. Before taking part in sports competition, athletes often strive to gain the lowest possible body mass in order to qualify for a lower weight category in relation to their individual potential depending on body build [9]. In combat sports, athletes often dehydrate to quickly reduce weight during a pre-tournament weigh-in [10].

A K1 kickboxing fight has been reported to induce high physiological stress [11]. Kickboxers are prone to injuries, especially to the head and neck [12]. A longer kickboxing practice creates a risk of pituitary hormone-secretion impairment (hypopituitarism), which manifests itself in decreased concentrations of growth hormone, ACTH, and IGF-I [13]. A single fight in kickboxing, just as in boxing, is an effort in which anaerobic and glycolytic processes are the source of energy, which is indicated by the high concentration of blood lactate [14,15]. Apart from the significant acidification after a kickboxing fight, large changes in hormone concentrations, especially cortisol and growth hormone, similar in winning and losing competitors, have also been reported [16]. During exercise, about 75% of the energy expended is heat and 25% is mechanical work. As a result, mechanical work is accompanied by an increase in body temperature, which depends on the intensity, duration of work, and external conditions and the possibility of heat release into the environment. The energy expended for heat production during the competitive effort has been estimated in athletes of different sports. In boxers, the level of this energy exceeds 1000 watts [17]. The heat exchange during a short but very intensive effort such as a kickboxing bout can be one of the determinants of the level of cognitive abilities and, consequently, the course of the fight. For this reason, it may, therefore, be important to determine skin temperature and pH during kickboxing bouts, as these parameters have not yet been determined for kickboxing in the K1 style. The results of the examination may provide answers regarding the cutaneous changes and body responses caused by the kickboxing bout and the related punches and kicks. The human body is warm-blooded and, through thermoregulatory processes, maintains a constant temperature regardless of changes in the ambient temperature [18,19]. Maintaining a relatively constant body core temperature is a prerequisite for the efficient functioning of the organs, including the activity of enzymes that control metabolism. The most constant temperature is in the right ventricle, while the highest temperature, apart from the heart, is in the liver, brain, and brown adipose tissue [20]. The thermal balance in skeletal muscles varies over a very wide range, with the amount of heat released increasing during work and decreasing at rest [21]. At rest, muscles generate heat depending on their contraction maintained by nerve impulses [22]. Skin temperature (ST) varies over a wide range on the body surface, especially in cold environments and during exercise [23]. Hot water, surfactants, or mechanical actions lead to the damage to the skin’s protective barrier or disturbance of the pH level [24]. The skin is protected and, therefore, it can effectively protect the inside of the human body when its pH remains at its natural level, allowing for a healthy balance of bacterial flora to be maintained [25].

The pH of the skin ranges between 4.5 and 5.5 [26]. In resting conditions, it was around two, but increased up to above five as a response to exercise [27,28]. In contrast, blood pH values are always much higher, even following intensive exercise. This is due to the buffer system that neutralizes the effects of lactic acidosis [28]. An alkaline pH facilitates the spread of bacteria throughout the epidermis, promotes the growth of pathogenic strains, and encourages the growth of bacteria that contribute to the unpleasant smell of sweat [29]. An acidic pH helps to keep the number of microorganisms at an appropriate level, increases the activity of bactericidal proteins and lipids, and facilitates proper skin keratinization and exfoliation, and wound healing [30]. Therefore, it is important to determine the local temperature at specific locations on the body surface and the pH level. An interesting issue is also the search for relations between skin temperature and its pH and the values of indicators of technical preparation.

A review of the literature on body composition and physiological variables in martial arts and combat sports reveals a lack of comprehensive analyses of body composition and cutaneous responses during fights in martial arts and combat sports. Research is often devoted to the aspects of proper hydration and the consequences of dehydration in sports [31]. The limits of dehydration and its consequences have also been explored [32]. Silva et al. [33] determined the body composition and power changes in elite judo athletes and confirmed the important role of proper hydration in judo competitions. The anthropometric characteristics and body composition of karate players at different sports skill levels have been also determined [34]. Scientific research on kickboxing has been mainly concerned with physiological and biochemical aspects [35,36,37]. Partial time-motion analyses of a kickboxing fight have also been conducted [38]. Previous research suggests that in addition to body mass, body composition may also be related to performance during athletic competition [39]. Additionally, the level of technical skills can be affected, to some extent, by the physique and body proportions of the athletes. For example, athletes with long limbs gain an advantage over their rivals by increasing the range of their attacks [6,40]. The literature review revealed the lack of studies in the context of measuring skin temperature and acidity during kickboxing competitions and linking body composition to indices of technical and tactical skill levels. 

Thus, the present study aimed to verify the body composition of athletes, their skin temperature and its acidity during a kickboxing K1 rules competition and also the level of the technical and tactical skills of the athletes in relation to body composition, weight category, and skin pH and temperature. We hypothesized that the temperature and pH of the skin change with each round of combat as a result of direct skin contact with blows delivered by the opponent. 

## 2. Materials and Methods

### 2.1. Participants

Twenty-four male kickboxers (Age range: 19 to 28 years), from different weight classes −71, −75, −81, −86, −91, and +91 kg, competing in a K1 kickboxing league volunteered to participate in the study. The inclusion criterion was primarily the sports’ skill level, which was determined based on training experience of at least 6 years including 4 years of active participation in competitions and informed consent to participate in the study. A detailed description of the participants divided into weight categories is presented in Table 1. The athletes were participating regularly in kickboxing tournaments for more than 2 years. The rank of the tournaments is varied. The athletes compete both at a low level (category C), a middle level (category B), and a high level (category A). For the purpose of this study, category A athletes were analyzed. They were also all assigned to the same training regimen four times per week (1.5 h per session). Athletes did not present any medical restrictions during the experimental period and refrained from any strenuous exercises for 48 h before the experimental sessions started. Furthermore, subjects were not advised to follow a special diet and were asked to refrain from all forms of additional supplementation. The dietary assessment of the experimental group was based on the interview method. The respondents kept records in a notebook where they noted the foods, dishes, and drinks consumed on a daily basis. Without weighing, they recorded portion sizes using home measures based on a photo album of produce and dishes provided. The recording procedure was performed for 3 days: 2 working days, 1 day off [41].

Analysis of dietary records showed no special diets or use of nutrients and supplements to enhance exercise in the training groups. Control of diet and supplementation helped exclude factors that could significantly interfere with the results of the experiment. The body mass of the participants ranged from 67.9 to 92.6 kg (mean: 81.3 ± 8.38 kg). Body fat percentage measured for the participants was estimated at a mean level of 14.09% and ranged from 5.8 to 27.0%. Furthermore, the level of muscle tissue of the participants ranged from 55.8 to 75.0 kg, with a mean of 65.12 kg. The mean bone mass was 3.46 kg, and its variation was found to be between 2.9 and 3.9 kg. Body mass index (BMI) of the participants ranged from 21.4 to 27.17 kg/m^2^, with a mean of 24.59 kg/m^2^. Body water content in the athletes studied was determined at a mean level of 63.46%, with the obtained values ranging from 56.1 to 69.4%.

Prior to participation in the tests, the competitors were informed about the research procedures, which were in accordance with the recent ethical principles of the Declaration of Helsinki (WMA, 2013). Athletes provided written informed consent after the explanation of the aims, benefits, and risks of the study. The research was approved by the Bioethics Committee at the Regional Medical Chamber (No. 287/KBL/OIL/2020).

### 2.2. Procedures

The examinations were conducted during a local kickboxing tournament that was refereed. The bouts were held based on K1 rules in accordance with the regulations of the World Association of Kickboxing Organizations (WAKO) and lasted for three rounds of 2 min each. The breaks between rounds lasted 1 min, and the measurements were made during this time (Figure 1). The tournament hall was equipped with an air conditioning system to keep a constant ambient temperature (20–21 °C) and humidity (50–52%) throughout the study.

### 2.3. Sparring Bout Analysis

The analysis of the sports fight was performed by two champion-level kickboxing coaches and one referee based on digital video recordings of the examined athlete. The recording was made with three cameras. Movavi Video Editor 14 software (Movavi, Wildwood, MO, USA) was used to merge the images. The setting of cameras allowed continuous observation of the athletes, referees, and the scoreboard. After the competition, the indices of attack activeness (A_a_), which represents the ratio of all offensive actions used during the fight, attack effectiveness (E_a_), which is the ratio of scoring techniques to all attacks used, and attack efficiency (S_a_), which is the number of attacks scored, were calculated using established equations from the literature [2,4,42].

Activeness of the attack (A_a_)
$$A_{a}=\frac{\mathrm{number}\;\mathrm{of}\;\mathrm{all}\;\mathrm{registered}\;\mathrm{offensive}\;\mathrm{actions}\;\mathrm{of}\;a\;\mathrm{kickboxer}}{\mathrm{number}\;\mathrm{of}\;\mathrm{bouts}\;\mathrm{fought}\;\mathrm{by}\;a\;\mathrm{kickboxer}}$$

Effectiveness of the attack (E_a_)
$$E_{a}=\frac{\mathrm{number}\;\mathrm{of}\;\mathrm{efective}\;\mathrm{attacks}}{\mathrm{number}\;\mathrm{of}\;\mathrm{all}\;\mathrm{attacks}}\;\times \;100$$

An effective attack is a technical action awarded a point.

Number of all attacks is a number of all offensive actions.

Efficiency of the attack (S_a_)
$$S_{a}=\frac{n}{N}$$

*n*—number of attacks awarded 1 pt. 

In K1 formula, each fair hit is awarded 1 pt.

*N*—number of bouts.

### 2.4. Biomedical Measurements

#### 2.4.1. Skin Temperature Measurement

Skin temperature was measured using a professional Skin-Thermometer ST 500 (Khazaka Electronic Germany) in degrees Celsius (°C). The measurement was performed using a special lens and an IR detector by measuring the infrared radiation (IR) emitted by the skin. Measurements were taken before the bout, and after the first, second, and third rounds. The measurement site was previously wiped with a dry towel. The single measurement time was 2 s.

#### 2.4.2. Skin/Sweat pH Measurement

Skin surface acidity was measured using the Skin-pH- Meter PH 905 (Khazaka Electronic Germany). Both skin temperature and pH measurements were taken at the following locations: forehead, chest, arm, hand, thigh, shank, and foot. After the measurements were completed for an individual, the probe was cleaned. The measurement time was one second.

#### 2.4.3. Heart Rate Measurement

Heart rate (HR) during a bout was also measured using a chest strap (Garmin HRM-PRO) and a specialized watch Garmin Fenix 6x pro (Garmin, Olathe, KS, USA). The strap was worn after each round of the fight to determine the heart rate at the end of the round.

### 2.5. Body Composition Analysis

Body composition was determined using the electrical bioimpedance technique using the Tanita Bc 601 body composition analyzer before the tournament, from 6 to 8 am during the weigh-in before the competition. Body mass, body fat, muscle tissue, bone mass, BMI, DCI, metabolic age, and body water content were determined. All parameters were automatically calculated using the measuring device.

### 2.6. Statistical Analysis

Descriptive statistics (mean, median, minimum and maximum values, first and third quartile values, and standard deviation) were calculated for all variables. Statistical analysis was performed using the statistical software package Statistica for Windows (version 13.1; Tulsa, OK, USA). The normality of data sets was checked and confirmed using the Shapiro–Wilk W test. The correlations between two variables with normal distribution (activeness of the attack, effectiveness of the attack, body mass, muscle tissue level, DCI) were determined using Pearson’s linear correlation coefficient, whereas for variables not meeting the criterion of normal distribution (efficiency of the attack, body fat %, bone mass, BMI, metabolic age, body water %), Spearman’s rank correlation coefficient was calculated. Changes in skin temperature, HR, and pH over time were evaluated using Friedman’s ANOVA. The level of statistical significance was set at *p* < 0.05. To control type-I error for multiple comparisons, Bonferroni procedure for correction of *p*-value has been used.

## 3. Results

The activeness of the attack in the athletes studied was estimated to range from 61 to 129 points, with a mean of 100.04 points. The efficiency of the attack was between 43 and 69 points, with a mean of 53.63 points. The effectiveness of the attack was at a mean level of 54.42 points, and it ranged from 40 to 69 points (Table 2).

Statistical analysis confirmed the presence of statistically significant relationships between the activeness, effectiveness, and efficiency of the attack, and most anthropometric characteristics of the athletes (Table 3). For other correlations, the higher activeness, efficiency, and effectiveness of the attack correlated with a lower body mass of athletes, lower body fat percentage, lower body fat, lower BMI, DCI, metabolic age, and higher body water percentage (all *p* < 0.05) (Table 3). There was an unexpected positive correlation between the effectiveness of attack and the variable obtained from the aggregation of the individual pH involving four time points and seven places on the body’s surface. This suggests that the more alkaline the skin is, the higher the index of effectiveness of attack is (Table 3).

Significant correlations were found between the selected anthropometric characteristics and the weight classes. There were positive relationships of weight class with body mass, body fat percentage, muscle tissue level, BMI, DCI, and metabolic age (Table 4). The value of the above-mentioned parameters increased in the athletes with the higher weight class. There was a statistically significant, negative correlation between the weight class and body water content in the athletes studied. Those performing in the higher weight classes had a lower body water percentage. Furthermore, there were also negative relationships of the activeness, efficiency, and effectiveness of the attack with weight class (Table 4).

The mean temperature dropped with each round of the bout on the chest, arm, and thigh. On the hand and foot, the skin temperature increased with each round (Table 5).

The highest pH was recorded on the forehead in the measurement after the third round of the bout, while the lowest was found for the thigh at baseline. The acidity on the arm, hand, thigh, and shin increased with each round of the bout. However, it decreased on the foot and forehead in the next two rounds (Table 6).

The HR values increased in the subsequent rounds of the fight, with its peak value of 184.63 bpm (Table 7).

Statistically significant correlations were demonstrated between skin temperature and pH on the chest after the first, second, and third rounds of the bout. Significant correlations were also shown after the first and second rounds on the arms. Single correlations also occurred on the shank after the second round of the bout and foot after the third round. Furthermore, numerous correlations were shown between the lower and upper limbs, chest, and arm (Table 8).

There was a statistically significant correlation between the pH of the forehead skin surface and HR after the first, second, and third rounds of the bout. The relationship between HR after the third round of the bout and the pH of the chest skin surface was also significant. There was a statistically significant correlation between the thigh temperature and HR at baseline.

## 4. Discussion

The purpose of this study was to comprehensively determine the body composition, skin temperature, and skin pH during a real kickboxing bout in K1 style and to establish the relationships of technical and tactical skills with body composition and weight classes. The results showed significant correlations between individual body composition parameters (body mass, body fat %, muscle tissue, BMI, DCI, metabolic age, body water %) and weight class that occurred in almost measured parameters. This result showed that the lower the weight class was, the lower the muscle and body fat was in the athletes. Similarly, lighter athletes were characterized by lower BMIs and DCIs. Kickboxing is a sport characterized by weight divisions where the competitor must meet certain body mass limits [6]. A negative correlation was found between the body water percentage and the weight class, which may also explain the limits associated with a specific weight class. Athletes often aim to compete in the lowest possible weight class by reducing their body mass and often inducing dehydration [43]. The present paper determined the level of technical and tactical skills of the athletes by analyzing their kickboxing fights according to K1 rules. Technical and tactical indices are the most precise tool used to determine athletes’ behavior in combat sports [4]. The statistical analysis showed a negative correlation between the weight classes and the level of technical and tactical skills of the athletes. This result can be explained by the fact that lighter competitors are characterized by greater dynamics during combat, with a greater variety and frequency of techniques used. Similar conclusions were reported by previous studies analyzing judo bouts, which highlighted technical variation in relation to specific weight classes [44,45]. Our results indicate differences in the skill used by weight category. They do not unequivocally indicate the level of these skills, but they can indicate the fact that depending on the weight category, different technical patterns dominate; lighter athletes fight faster using more techniques and heavier athletes use fewer techniques.

Our research showed that a low level of body fat and muscle tissue significantly correlates with the activeness, effectiveness, and efficiency of the attack. The body fat percentage measured among the kickboxers was estimated at a mean level of 14.09%, ranging from 5.8 to 27%. The body fat increase was connected with a heavier weight class, in which the athletes were characterized by a higher body mass and lower activeness during the fight, which translated into other performance indices (effectiveness and efficiency). Furthermore, the mean fat percentage in athletes in the present study showed an optimal range according to the accepted norms [46,47]. The body fat percentage found in the present study was similar to that reported in a previous study of boxers, where body fat ranged from 9 to 16% [48]. This may indicate a convergence in the desired low levels of body fat in representatives of both sports. A detailed analysis of the examination conducted on kickboxing athletes revealed that a low body fat percentage is a prerequisite for athletes’ high sports performance [6]. Likewise, athletes’ muscle tissue level in the present study was found to have an average of 65.12 kg. Muscle development is related to both genetic predisposition and the training process, which shapes mainly the leg, arm, and abdominal muscles [49]. However, scientific studies confirmed that contact sports’ competition induced significant muscle fatigue and damage [50]. Therefore, optimal muscle development is essential to obtain successful performance during combat, especially for technical and tactical actions [2].

Our study showed changes in skin temperature during the fights. Regular decreases in ST were found on the chest, arm, and thigh following each round of the kickboxing fight. The decrease in skin temperature in these areas may have been related to the evaporation of excess sweat, which is an endothermic process. Hence, sweating of the skin surface is considered the most effective way for the dissipation of excess heat in the human body that appears during prolonged and/or intensive exercise [51]. A decrease in temperature can also be caused by the changed blood flow due to compensatory vasoregulation [52,53]. In this consideration, Barboza et al. [34] assessed the skin temperature of middle-distance runners during maximal exercise and showed a decrease in the upper body area’s temperature, while it was increased in the upper limbs due to solicited muscles [54]. In the present study, increased skin temperature was found on the feet, which may be due to metabolic heat generation or stress [55]. During exercise, blood flow increases in order to oxygenate the tissues, and therefore, temperature may increase [56]. It should be stressed that strenuous exertion, an elevated body or ambient temperature are not the only causes of an increased rate of sweating. Strong emotions, fear, and so-called psychological stress are independent factors leading to sweating, as has been found in pianists prior and post their official performance [57].

The pH analysis of the skin showed an acidic reaction in each case. After the first round of the fight, the pH level relative to the previous measurement slightly decreased on the forehead, chest, and foot, but after the third round, it increased over the initial value. This behavior might by related to changes in chemical substances in sweat such as hydrogen ions donors, lactic acid, or their acceptors, such as ammonia [58,59]. Measurements of pH in other points of the body also showed an increase between the rounds and the resting value. Searching the other reasons for this phenomenon, it is worth emphasizing that the human body has two types of sweat glands (eccrine, apocrine), which produce sweat in different amounts, directly affecting the pH [59].

For heart rate measurements, our results showed that values increased in exercise between the baseline and the first round, which may suggest the presence of anaerobic glycolysis. Previous research reported that kickboxing fights caused substantial physiological stress [11]. Our results showed a negative correlation between skin temperature and skin pH on the chest and arm, which indicated an increase in skin temperature at lower pH values. Statistical analysis revealed a correlation between the heart rate after the first round and the pH of the skin on the forehead, which was increased. In the remaining rounds of the bout, the pH value for the forehead decreased with higher HR values, which can be explained by an anaerobic metabolism and high body acidification [11].

The increase in HR in successive rounds reflects the increase in activity of the autonomic nervous system (ANS). The same behavior of the HR was noted during the boxing fights [58]. This response of the cardiovascular system and, indirectly, the nervous system may have influenced changes in both temperature and pH in our study. The mechanism of these relationships has been discussed above.

### Limitation of the Study

In the present study, heart rate was measured by wearing a measuring strap after each round. Therefore, we were not able to record the maximum heart rate during the bout. The judges did not allow the strap to be worn during the entire bout for safety reasons. An additional difficulty during the examination was the profuse perspiration, and therefore, the need to wipe the examination site. Additionally, we did not have possibility to examine athletes from all the weight categories and female athletes.

## 5. Conclusions

Kickboxers who compete in lower weight classes are likely to be characterized by higher technical and tactical skills. The level of body fat and muscle tissue can affect the level of technical and tactical performance. The skin temperature changed with each round of the fight, and a temperature decline was noted in the large muscle groups (chest, arm, thigh) as the fight progressed. A kickboxing fight according to K1 rules led to skin pH changes after each round of the bout in the study group.

In conclusion, it should be emphasized that the directions of pH and temperature changes observed on the skin’s surface during exercise may be very different from those occurring inside the body that were well described in the literature. Advanced techniques for measuring physiological changes on the skin have only recently become available to researchers; therefore, the small number of similar experimental studies published to date do not fully explain the physiological mechanisms of the observed phenomena.

### Practical Implications

In kickboxing, body composition should be constantly monitored because the measured values can affect the course of the fight and the level of technical and tactical skills. Further research should also be conducted to clarify the physiological changes of the skin’s surface during combat.

## Figures and Tables

**Figure 1 ijerph-18-11625-f001:**
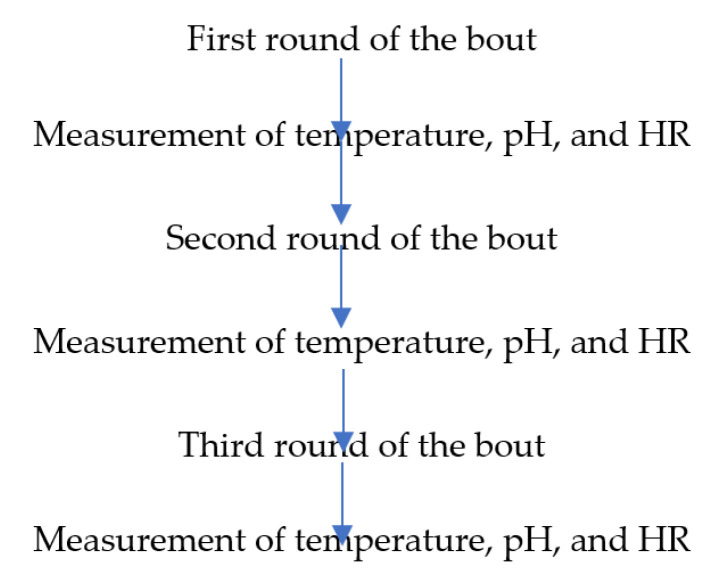
Research design.

**Table 1 ijerph-18-11625-t001:** Characteristics of athletes in each weight category.

Weight Classes of Athletes	N	Age	Body Mass	BMI	%
−71 kg	4	23.0 ± 4.24	68.9 ± 1.09	21.37 ± 0.21	16.7%
−75 kg	4	21.75 ± 2.75	73.9 ± 0.44	22.26 ± 0.49	16.7%
−81 kg	4	21.25 ± 2.21	78.92 ± 1.35	24.89 ± 0.53	16.7%
−86 kg	4	21.5 ± 2.65	85.05 ± 1.18	25.8 ± 0.29	16.7%
−91 kg	4	21.3 ± 2.21	89.0 ± 1.20	26.03 ± 0.32	16.7%
+91 kg	4	23.0 ± 3.74	91.92 ± 0.58	26.58 ± 1.20	16.7%
Total	24	21.95 ± 2.82	81.28 ± 8.39	25.99 ± 2.04	100.0%

N—number of observations; %—percentage. Source: author’s own elaboration.

**Table 2 ijerph-18-11625-t002:** Level of indices of technical and tactical skills of athletes.

Index	Descriptive Statistics							
	*N*	Mean	Med.	Min.	Max.	Lower Quartile	Upper Quartile	Std. Deviation
Activeness of the attack	24	100.04	100.50	61.00	129.00	89.50	116.50	18.36
Efficiency of the attack	24	53.63	50.50	43.00	69.00	46.50	60.50	8.07
Effectiveness of the attack	24	54.42	56.00	40.00	69.00	47.50	59.50	8.09

*N*—number of observations; $\bar{x}$—arithmetic mean; Me—median; Min—minimum; Max—maximum; Q1—lower quartile; Q3—lower quartile; SD—standard deviation.

**Table 3 ijerph-18-11625-t003:** Evaluation of relationships between indices of technical and tactical skills and anthropometric characteristics (*n* = 24).

Variables	Activeness of the Attack		Efficiency of the Attack		Effectiveness of the Attack	
	r/R	*p*	r/R	*p*	r/R	*p*
Body mass	**−0.88**	**<0.001**	**−0.87**	**<0.001**	**−0.82**	**<0.001**
Body fat %	**−0.67**	**<0.001**	**−0.80**	**<0.001**	**−0.75**	**<0.001**
Muscle tissue	**−0.80**	**<0.001**	**−0.66**	**<0.001**	**−0.60**	**0.002**
Bone mass	−0.33	0.110	−0.13	0.556	−0.16	0.462
BMI	**−0.94**	**<0.001**	**−0.83**	**<0.001**	**−0.75**	**<0.001**
DCI	**−0.59**	**0.003**	−0.27	0.195	**−0.45**	**0.029**
Metabolic age	**−0.62**	**0.001**	**−0.66**	**<0.001**	**−0.67**	**<0.001**
Body water content (%)	**0.61**	**0.001**	**0.69**	**<0.001**	**0.65**	**0.001**
Mean skin pH	0.28	0.284	**0.42**	**0.039**	**0.43**	**0.043**
Mean skin temperature	0.17	0.424	0.14	0.051	0.5	0.813

BMI: body mass index, DCI: daily calorie intake, r—Pearson linear correlation; R—Spearman’s rank correlation; *p*—test probability; values in bold are statistically significant. Source: author’s own elaboration.

**Table 4 ijerph-18-11625-t004:** Evaluation of the relationship of effectiveness and anthropometric characteristics with weight class (*n* = 24).

Variables	R	*p*
Body mass vs. weight class	**0.99**	**<0.001**
Body fat % vs. weight class	**0.78**	**<0.001**
Muscle tissue vs. weight class	**0.73**	**<0.001**
Bone mass vs. weight class	0.26	0.218
BMI vs. weight class	**0.93**	**<0.001**
DCI vs. weight class	**0.53**	**0.008**
Metabolic age vs. weight class	**0.74**	**<0.001**
Body water % vs. weight class	**−0.73**	**<0.001**
Activeness of the attack vs. weight class	**−0.94**	**<0.001**
Efficiency of the attack vs. weight class	**−0.87**	**<0.001**
Effectiveness of the attack vs. weight class	**−0.84**	**<0.001**

R—Spearman’s rank correlation; *p*—test probability; values in bold are statistically significant. Source: author’s own elaboration.

**Table 5 ijerph-18-11625-t005:** Descriptive statistics for temperature after warming up (WU) and three successive rounds.

Temperature	*N*	Mean	Confidence: −95%	Confidence: +95%	Med.	Min.	Max.	Lower Quartile	Upper Quartile	Std. Deviation
Forehead (WU)	24	31.92	31.35	32.49	32.50	30.00	33.40	30.00	32.99	1.34
1	24	30.52	29.87	31.16	30.60	27.40	32.80	29.70	31.00	1.53
2	24	30.73	30.11	31.34	31.40	28.30	32.30	29.40	32.10	1.45
3	24	31.13	30.68	31.58	31.10	29.20	32.60	30.45	31.80	1.07
*p*	**Chi^2^Anova = 20.35, *p* < 0.001, η square = 0.988**									
Chest (WU)	24	29.53	29.07	29.98	30.00	27.40	31.00	28.85	30.20	1.09
1	24	27.34	26.92	27.75	27.80	25.40	28.60	26.75	27.90	0.98
2	24	26.38	26.08	26.67	26.70	25.10	27.30	25.90	26.80	0.70
3	24	25.85	25.29	26.41	26.00	24.10	28.00	24.20	27.00	1.32
*p*	**Chi^2^Anova = 56.60 *p* < 0.001, η square = 0.998**									
Arm (WU)	24	29.23	28.86	29.60	29.50	27.70	30.40	28.55	30.00	0.88
1	24	27.85	27.31	28.38	28.10	26.10	29.80	26.30	28.60	1.27
2	24	26.96	26.31	27.61	27.30	24.70	28.80	25.20	28.70	1.54
3	24	26.71	26.23	27.19	26.70	24.70	28.10	26.00	28.10	1.13
*p*	**Chi^2^Anova = 53.69 *p* < 0.001, η square = 0.987**									
Hand (WU)	24	30.17	29.43	30.91	30.40	28.00	32.60	28.10	31.90	1.75
1	24	30.23	29.52	30.94	31.50	28.10	31.90	28.60	31.80	1.68
2	24	30.73	30.12	31.35	31.50	28.90	32.40	29.20	32.30	1.45
3	24	31.41	30.75	32.08	30.70	29.70	34.10	30.00	32.70	1.57
*p*	**Chi^2^Anova = 28.29 *p* < 0.001, η square = 0.991**									
Thigh (WU)	24	29.58	29.25	29.92	29.70	28.20	30.70	29.00	30.20	0.79
1	24	27.78	27.38	28.18	27.90	25.80	28.90	27.30	28.70	0.95
2	24	27.34	26.72	27.95	27.30	25.20	30.00	26.10	27.60	1.45
3	24	27.06	26.44	27.68	26.80	25.30	29.50	25.70	28.20	1.46
*p*	**Chi^2^Anova = 49.05 *p* < 0.001, η square = 0.997**									
Shank (WU)	24	29.14	28.86	29.43	29.20	27.70	30.10	29.00	29.50	0.67
1	24	27.68	27.36	28.01	27.90	26.30	28.90	27.20	28.20	0.77
2	24	27.10	26.78	27.43	27.00	25.50	28.10	26.90	27.75	0.77
3	24	27.21	26.91	27.52	27.20	25.80	28.20	26.80	28.00	0.73
*p*	**Chi^2^Anova = 57.88 *p* < 0.001, η square = 0.999**									
Foot (WU)	24	24.98	24.03	25.92	24.00	23.10	29.70	23.20	26.90	2.24
1	24	25.06	24.19	25.93	25.00	23.00	29.60	23.35	25.90	2.07
2	24	25.74	24.54	26.93	26.00	21.30	30.00	23.20	27.60	2.83
3	24	26.03	25.05	27.00	25.30	22.50	29.50	24.90	28.50	2.30
*p*	Chi^2^Anova = 6.85 *p* = 0.077, η square = 0.108									

*p*—test probability; values in bold are statistically significant η square-effect size.

**Table 6 ijerph-18-11625-t006:** Descriptive statistics for pH after WU and following successive rounds.

pH	*N*	Mean	Confidence: −95	Confidence: +95%	Med.	Min.	Max.	Lower Quartile	Upper Quartile	Std. Deviation
forehead (WU)	24	6.23	6.13	6.33	6.22	5.80	6.48	6.09	6.43	0.24
1	24	5.86	5.73	5.99	5.92	5.40	6.28	5.61	6.13	0.31
2	24	6.11	5.91	6.31	5.96	5.35	6.79	5.83	6.62	0.47
3	24	6.55	6.37	6.72	6.69	5.92	7.02	6.01	6.90	0.42
*p*	**Chi^2^Anova = 30.55 *p* < 0.001, η square = 0.992**									
chest (WU)	24	5.81	5.74	5.88	5.87	5.51	5.97	5.68	5.96	0.17
1	24	5.65	5.43	5.86	5.83	4.91	6.30	5.10	6.11	0.51
2	24	5.90	5.64	6.16	5.92	4.93	7.01	5.38	6.04	0.62
3	24	6.20	5.97	6.44	6.21	5.31	7.16	5.76	6.52	0.57
*p*	**Chi^2^Anova = 36.45 *p* < 0.001, η square = 0.993**									
arm (WU)	24	5.58	5.44	5.72	5.69	4.89	5.89	5.32	5.89	0.33
1	24	5.68	5.44	5.93	6.03	4.83	6.27	5.11	6.22	0.57
2	24	5.86	5.54	6.18	5.91	4.84	7.17	5.25	6.16	0.75
3	24	6.13	5.81	6.44	6.02	5.14	7.55	5.66	6.32	0.75
*p*	**Chi^2^Anova = 40.85 *p* < 0.001, η square = 0.994**									
hand (WU)	24	5.99	5.86	6.11	6.01	5.62	6.44	5.69	6.23	0.30
1	24	6.07	5.87	6.27	6.11	5.39	6.80	5.66	6.42	0.48
2	24	6.12	5.84	6.40	6.19	5.21	7.21	5.59	6.55	0.66
3	24	6.24	6.00	6.48	6.08	5.53	7.26	5.81	6.47	0.57
*p*	**Chi^2^Anova = 13.85 *p* = 0.003, η square = 0.388**									
thigh (WU)	24	5.49	5.37	5.61	5.47	5.13	5.90	5.22	5.78	0.27
1	24	5.66	5.40	5.93	5.77	4.87	6.73	5.04	5.91	0.63
2	24	5.74	5.42	6.06	5.58	4.87	7.12	5.11	6.04	0.76
3	24	5.85	5.50	6.20	5.79	4.85	7.38	5.21	6.05	0.83
*p*	**Chi^2^Anova = 11.25 *p* = 0.010, η square = 0.319**									
shank (WU)	24	5.55	5.42	5.67	5.59	5.11	5.93	5.34	5.78	0.29
1	24	5.57	5.33	5.82	5.62	4.51	6.40	5.26	5.82	0.58
2	24	5.66	5.28	6.05	5.43	4.67	7.43	5.00	6.04	0.91
3	24	5.86	5.54	6.17	5.67	4.95	7.17	5.25	6.31	0.75
*p*	**Chi^2^Anova = 12.10 *p* = 0.007, η square = 0.330**									
foot (WU)	24	5.91	5.71	6.12	5.68	5.52	6.87	5.62	6.20	0.48
1	24	5.66	5.41	5.90	5.54	4.99	6.83	5.28	5.72	0.58
2	24	5.74	5.50	5.99	5.47	5.07	6.82	5.29	6.03	0.58
3	24	6.11	5.78	6.44	5.92	5.25	7.63	5.57	6.36	0.78
*p*	**Chi^2^Anova = 38.45 *p* < 0.001, η square = 0.993**									

*p*—test probability; values in bold are statistically significant.

**Table 7 ijerph-18-11625-t007:** Descriptive statistics for heart rate (HR) measurements following warm-up (WU) and successive rounds.

HR	*N*	Mean	Confidence: −95%	Confidence: +95%	Med.	Min.	Max.	Lower Quartile	Upper Quartile	Std. Deviation
WU	24	116.96	111.13	122.79	117.00	103.00	148.00	104.00	119.50	13.81
First round	24	179.50	177.03	181.97	181.00	172.00	190.00	174.00	183.00	5.85
Second round	24	183.33	181.20	185.46	185.00	175.00	190.00	179.00	187.00	5.04
Third round	24	184.63	182.03	187.22	188.00	174.00	191.00	181.00	190.00	6.13
*p*	**Chi^2^Anova = 59.27 *p* < 0.001, η square = 0.999**									

*p*—test probability; values in bold are statistically significant.

**Table 8 ijerph-18-11625-t008:** Relationship between pH and temperature (*n* = 24).

		Temperature						
		Forehead	Chest	Arm	Hand	Thigh	Shank	Foot
**pH**	forehead (WU)	**−0.82**	**−0.60**	−0.24	**−0.51**	**−0.73**	−0.22	−0.16
	1	0.28	**−0.45**	**−0.68**	0.40	**−0.57**	−0.02	−0.31
	2	0.19	**−0.93**	**−0.73**	**0.49**	**−0.42**	−0.33	**0.53**
	3	−0.16	0.05	**−0.76**	0.24	**−0.50**	−0.18	0.21
	chest (WU)	0.39	0.14	−0.32	0.01	**0.55**	−0.22	−0.31
	1	**0.46**	**−0.45**	**−0.68**	0.33	**−0.57**	−0.28	−0.13
	2	0.40	**−0.97**	**−0.83**	0.39	**−0.53**	−0.39	**0.46**
	3	0.17	**−0.51**	**−0.68**	**0.45**	**−0.54**	0.05	**0.55**
	arm (WU)	**0.50**	0.22	−0.07	−0.16	0.35	**−0.41**	**−0.50**
	1	**0.54**	−0.20	**−0.43**	0.25	−**0.42**	−0.29	−0.13
	2	**0.77**	**−0.62**	**−0.43**	0.01	−0.34	**−0.59**	0.19
	3	**0.65**	**−0.87**	−0.16	**0.42**	−0.25	−0.24	0.18
	hand (WU)	**0.72**	**0.70**	**0.54**	0.34	**0.43**	0.29	0.07
	1	**0.64**	−0.22	**−0.57**	0.40	**−0.71**	**−0.51**	−0.35
	2	**0.62**	**−0.67**	**−0.70**	0.06	**−0.68**	**−0.68**	−0.02
	3	**0.70**	**−0.96**	−0.24	**0.53**	**−0.44**	**−0.48**	−0.04
	thigh (WU)	**0.92**	0.17	0.15	**0.52**	−0.02	−0.12	0.25
	1	**0.52**	−0.12	−0.25	0.07	−0.16	−0.31	0.09
	2	**0.66**	**−0.44**	−0.32	−0.09	−0.36	**−0.45**	0.04
	3	0.35	**−0.78**	−0.18	0.27	−0.33	−0.22	0.05
	shank (WU)	**0.97**	0.16	0.05	**0.65**	0.13	−0.11	0.38
	1	0.15	0.02	−0.07	0.04	**−0.45**	−0.37	−0.25
	2	**0.60**	**−0.58**	**−0.43**	−0.11	−0.39	**−0.43**	0.13
	3	**0.43**	**−0.83**	−0.05	0.35	−0.31	−0.27	−0.04
	foot (WU)	**0.78**	0.04	0.16	0.41	0.40	−0.36	0.16
	1	0.04	−0.32	**−0.43**	0.16	**−0.81**	**−0.48**	−0.40
	2	**0.58**	**−0.82**	**−0.77**	0.13	**−0.63**	**−0.51**	0.20
	3	0.31	**−0.78**	**−0.41**	0.32	**−0.64**	**−0.75**	**−0.41**

Values in bold are statistically significant.

## Data Availability

The data presented in this study are available on request from the corresponding author.

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
