# Peer review of "Evaluation of the Body Composition and Selected Physiological Variables of the Skin Surface Depending on Technical and Tactical Skills of Kickboxing Athletes in K1 Style"

_ijerph, 2021, doi:10.3390/ijerph182111625_

Round 1
Reviewer 1 Report
To Mr Rydzik and colleagues,
Thank you for submitting this novel study for review. I believe this is an interesting study with some potentially useful findings for the preparation and safety of combat sports athletes. There are several key improvements required before this manuscript is ready for publication. Please review the following comments and suggestions in editing and improving your work.
Kind regards,
Christopher Kirk
Abstract
Please be cautious about stating that one variable (body fat for example) has affected another (technical/tactical skills for example), as you have only shown that these are correlated. You have not demonstrated a definitive link or causative relationship between the two. Also, the statement “skin temperature decreased with each round of the bout” is not true for all body regions reported and seems to be dependent on whether the mean or the median is being considered.
Introduction
Lines 28-35: I feel the authors could provide a more instructive description of what K1 athletes are required to do in competition. Which techniques are allowed/prohibited in relation to other kickboxing styles? Is muay Thai being included here, as this would appear to have fewer restrictions on techniques allowed in competition (time and number of strikes allowed in the clinch for instance). This section is key for non-combat sports reader/researchers.
Lines 35-42: Please use ‘body mass’ in place of ‘weight’ in keeping with correct science terminology. Also, how strong is the evidence that differences in body-limb ratios has an effect on performance or success in combat sports? Do different limb lengths improve or have no effect on success?
Lines 46-49: The weight class of combat sports athletes is highly influenced by their weight cutting practices prior to competition so I feel this needs introducing here, especially with your study measuring body water and body fat as dependent variables.
Lines 51-68: I think this section includes a lot of information not directly relevant to the nature and content of this study. I think this section needs to be more focussed around the specific physiological effects of kickboxing/K1 in terms of O2 consumption, lactate production and any information that already exists regarding the temperature and sweat responses to kickboxing or other related combat sports.
Lines 69-70: I think this section needs to be more focused as well. It’s not clear how the discussion of physiological maintenance of blood pH is relevant to the measurement of skin pH during sports performance. Also, please remove any reference of lactate causing changes in pH, as this is not the case. Changes in blood pH are caused by CO2 and H+, not lactate.
Lines 87-90: Please provide some evidence that body composition might affect technical/tactical skills in competition. As this forms a large portion of your study, the introduction needs to justify its inclusion.
Lines 93-95: Why would ‘numerous blow’ alter temperature or pH? Is this the number of strikes thrown or received by the participant?
Methods
Lines 99-100: Please report how many participants were in each group as well and the mean age, body mass and stature of each weight class group.
Line 100-102: Please define what is meant by ‘local’ and ‘regional’ tournaments in this context. Are the participants competing at a low, intermediate or high level?
Lines 105-106: Please provide more details of this process and what the purpose of this was in the context of the study. What was the aim of the nutritional programs? Who provided this? Why? What were the interviews conducted for? Why?
Lines 106-116: How were each of these measured? As dependent variables this needs describing in detail – including methods and equipment used for each variable mentioned. Metabolic age especially requires a detailed explanation as to how this was determined and what it means in practice. Although the Tanita Bc 601 is mentioned in line 172-173, this is not enough information to allow replication or understanding of the procedures employed.
Lines 145-146: Please provide brief definitions of each of these variables as key dependent variables of your study.
Lines 158-161: Please provide manufacturer details of the equipment used. How long did each measurement take and what specific units were measured and reported?
Lines 164-166: Please provide manufacturer details of the equipment used. How long did each measurement take and what specific units were measured and reported?
Lines 168-170: Was the strap worn during the bouts or was it placed on between rounds? How long did this process take and was the HR monitor lag time taken into account in the analyses?
Lines 172-173: When was this performed?
Lines 174-176: Why were only medallists included? How many bouts were recorded per participant?
Lines 186-189: Please provide effect sizes for each ANOVA conducted to allow for future meta analyses.
Results
Table 1: What units are being reported here? Please be consistent between tables – x is used here to report the mean, with ‘Mean’ being used in all other tables.
Lines 213-220 and Table 3: The relationships between body composition and weight class were not mentioned as an aim or hypothesis of the study in the introduction. Equally, is this not to be expected as a result of people being of different body sizes anyway? Please justify the inclusion of these analyses in the introduction.
Lines 226-227: This statement is not accurate. When reported as the mean, some body region skin temperatures increase. When reported as the median, some body region temperatures increase from round 1 to round 2 before reducing in round 3. Please rephrase this to be more accurate in light of the reported data. Please also decide whether the mean or the median is more relevant for the data reported and determine whether both actually need to be reported.
Table 5: Please state which units pH has been recorded and reported in.
Discussion
Lines 270-272: Please ensure the stated aims in introduction match those discussed here.
Lines 284-289: Do these results indicate higher/lower technical skill? Or do they indicate that different weight classes display different patterns of technique use?
Line 310: If skin temperature requires abbreviating, then please do this the first time the term is used, not in the discussion.
Lines 310-326: Please refer to my previous comments regarding Lines 226-227, as this discusses skin temperature decreasing, which is not strictly true according to the data reported. Also, sweat has not been measured here, so please change the content of this section to discuss skin temperature rather than sweat.
Lines 327-331: Increased lactate does not lower blood pH. Please remove this statement, or alter it to discuss changes in CO2 and H+. Also, this section should discuss how the potential changes in blood pH affect the changes in skin pH that have been measured and how this relates to the activity observed in the bouts.
Lines 341-343: this requires more discussion – how do we know the participants were experiencing anaerobic metabolism?
Lines 344-345: Please expand on this and discuss it’s relevance to your data.
Conclusions
Please avoid using bullet points to conclude your manuscripts. Please also avoid making firm statements about causality using data that is correlational only.
Lines 356-358: is 14% a low body fat? What is an ‘optimal level of muscle’ and does your data show this? And as discussed above, correlational data cannot show that higher/lower muscle mass increases technical skill, please rephrase.
Author Response
Dear Reviewer,
Thank you very much for your time and valuable comments, which all have been considered and incorporated. The detailed list of responses is given below. We hope that the modifications and explanation will be acceptable for you.
Yours sincerely,
Rydzik, corresponding author
Abstract
Please be cautious about stating that one variable (body fat for example) has affected another (technical/tactical skills for example), as you have only shown that these are correlated. You have not demonstrated a definitive link or causative relationship between the two. Also, the statement “skin temperature decreased with each round of the bout” is not true for all body regions reported and seems to be dependent on whether the mean or the median is being considered.
A: This part has been corrected
Introduction
Lines 28-35: I feel the authors could provide a more instructive description of what K1 athletes are required to do in competition. Which techniques are allowed/prohibited in relation to other kickboxing styles? Is muay Thai being included here, as this would appear to have fewer restrictions on techniques allowed in competition (time and number of strikes allowed in the clinch for instance). This section is key for non-combat sports reader/researchers.
A: This has been corrected, Muay Thai was not included in the considerations due to the differences in the course of the fight. You are right that the time of the fight is similar. However K1 does not allow elbow strikes, multiple knee kicks in one clinch and throws.
Lines 35-42: Please use ‘body mass’ in place of ‘weight’ in keeping with correct science terminology. Also, how strong is the evidence that differences in body-limb ratios has an effect on performance or success in combat sports? Do different limb lengths improve or have no effect on success?
A: This part has been corrected
Lines 46-49: The weight class of combat sports athletes is highly influenced by their weight cutting practices prior to competition so I feel this needs introducing here, especially with your study measuring body water and body fat as dependent variables.
A: This part has been corrected
Lines 51-68: I think this section includes a lot of information not directly relevant to the nature and content of this study. I think this section needs to be more focussed around the specific physiological effects of kickboxing/K1 in terms of O2 consumption, lactate production and any information that already exists regarding the temperature and sweat responses to kickboxing or other related combat sports.
A: This part has been corrected
Lines 69-70: I think this section needs to be more focused as well. It’s not clear how the discussion of physiological maintenance of blood pH is relevant to the measurement of skin pH during sports performance. Also, please remove any reference of lactate causing changes in pH, as this is not the case. Changes in blood pH are caused by CO2 and H+, not lactate.
A:This sentence is not a discussion, it is an introduction in which we refer to general responses in the human body relating to pH changes caused by various factors including physical activity. One sentence was removed, there are not many studies that refer to pH and temperature measurements during real combat so our work is novel and we used general knowledge.
Lines 87-90: Please provide some evidence that body composition might affect technical/tactical skills in competition. As this forms a large portion of your study, the introduction needs to justify its inclusion.
A: This part has been corrected
Lines 93-95: Why would ‘numerous blow’ alter temperature or pH? Is this the number of strikes thrown or received by the participant?
A: This part has been corrected
Methods
Lines 99-100: Please report how many participants were in each group as well and the mean age, body mass and stature of each weight class group.
A: Tables have been added
Line 100-102: Please define what is meant by ‘local’ and ‘regional’ tournaments in this context. Are the participants competing at a low, intermediate or high level?
A: This part has been corrected
Lines 105-106: Please provide more details of this process and what the purpose of this was in the context of the study. What was the aim of the nutritional programs? Who provided this? Why? What were the interviews conducted for? Why?
A: A description was extended to explain all the details
Lines 106-116: How were each of these measured? As dependent variables this needs describing in detail – including methods and equipment used for each variable mentioned. Metabolic age especially requires a detailed explanation as to how this was determined and what it means in practice. Although the Tanita Bc 601 is mentioned in line 172-173, this is not enough information to allow replication or understanding of the procedures employed.
A: Description of body composition measurement technique has been added. Furthermore, all the information (metabolic age) was obtained from Tanita
Lines 145-146: Please provide brief definitions of each of these variables as key dependent variables of your study.
A: More information has been added
Lines 158-161: Please provide manufacturer details of the equipment used. How long did each measurement take and what specific units were measured and reported?
A: This part has been corrected
Lines 164-166: Please provide manufacturer details of the equipment used. How long did each measurement take and what specific units were measured and reported?
A: This part has been corrected
Lines 168-170: Was the strap worn during the bouts or was it placed on between rounds? How long did this process take and was the HR monitor lag time taken into account in the analyses?
A: The strap was worn after each round. The judges refused to allow the strap to be worn during the fight for safety reasons. We did not account for monitor latency. We have added this information in the Limitation of the study section
Lines 172-173: When was this performed?
A: This part has been corrected
Lines 174-176: Why were only medallists included? How many bouts were recorded per participant?
A: This has been corrected, it was a mistake
Lines 186-189: Please provide effect sizes for each ANOVA conducted to allow for future meta analyses.
A: Eta-squared effect size has been added.
Results
Table 1: What units are being reported here? Please be consistent between tables – x is used here to report the mean, with ‘Mean’ being used in all other tables.
A: This has been added as a bulleted list. The tables have been corrected
Lines 213-220 and Table 3: The relationships between body composition and weight class were not mentioned as an aim or hypothesis of the study in the introduction. Equally, is this not to be expected as a result of people being of different body sizes anyway? Please justify the inclusion of these analyses in the introduction.
A: Information has been added in the study aim regarding body composition. It seems to us that the relationship between body composition and weight category is interesting due to the fact that classification into a given category is based only on weight measurement. From a coaching point of view, body composition is also important because athletes need to have high-level lean body mass because fat tissue is considered unnecessary.
Lines 226-227: This statement is not accurate. When reported as the mean, some body region skin temperatures increase. When reported as the median, some body region temperatures increase from round 1 to round 2 before reducing in round 3. Please rephrase this to be more accurate in light of the reported data. Please also decide whether the mean or the median is more relevant for the data reported and determine whether both actually need to be reported.
A: This part has been corrected
Table 5: Please state which units pH has been recorded and reported in.
A: Ph is a dimensionless unit and is comparative in nature and does not translate directly into the concentration of hydronium Ions or any other
Discussion
Lines 270-272: Please ensure the stated aims in introduction match those discussed here.
A: The aim has been corrected.
Lines 284-289: Do these results indicate higher/lower technical skill? Or do they indicate that different weight classes display different patterns of technique use?
A: Our results indicate differences in skill use by weight category. They do not indicate unequivocally the level of these skills, but they can indicate the fact that depending on the weight category, different technical patterns dominate: lighter athletes fight faster using more techniques and heavier athletes use fewer techniques.
Line 310: If skin temperature requires abbreviating, then please do this the first time the term is used, not in the discussion.
A: This part has been corrected
Lines 310-326: Please refer to my previous comments regarding Lines 226-227, as this discusses skin temperature decreasing, which is not strictly true according to the data reported. Also, sweat has not been measured here, so please change the content of this section to discuss skin temperature rather than sweat.
A: In the first part of the discussion, we analyzed those parts of the body in which the temperature gradient occurred. In fact, we removed the shank because we agree with the Reviewer that there was no decline here. Thank you very much. Chest, arm, and thigh are large muscles and we tried to explain the temperature changes by sweating (which cooled the skin) based on the available literature. We didn't measure sweat, but it is noticeable during the fight. Based on the literature, we explained this phenomenon for discussing the causes of temperature changes.
Lines 327-331: Increased lactate does not lower blood pH. Please remove this statement, or alter it to discuss changes in CO2 and H+. Also, this section should discuss how the potential changes in blood pH affect the changes in skin pH that have been measured and how this relates to the activity observed in the bouts.
A: This part has been corrected
Lines 341-343: this requires more discussion – how do we know the participants were experiencing anaerobic metabolism?
A: In our study, we evaluated physiological mechanisms during a real kickboxing bout in K1 style. The published results showed that after round 3 the athletes had a heart rate value of 185 bpm with an LA concentration of 14.6mmol/l. In the current work, HR verification after the 3rd round of the duel showed similar values. Therefore anaerobic metabolism can be presumed.
Rydzik, Ł.; Maciejczyk, M.; Czarny, W.; Kędra, A.; Ambroży, T. Physiological Responses and Bout Analysis in Elite Kickboxers During International K1 Competitions. Front. Physiol. 2021, 12, 737–741, doi:10.3389/fphys.2021.691028.
Lines 344-345: Please expand on this and discuss it’s relevance to your data.
A: This part has been corrected
Conclusions
Please avoid using bullet points to conclude your manuscripts. Please also avoid making firm statements about causality using data that is correlational only.
A: The Conclusion section has been corrected
Lines 356-358: is 14% a low body fat? What is an ‘optimal level of muscle’ and does your data show this? And as discussed above, correlational data cannot show that higher/lower muscle mass increases technical skill, please rephrase.
A: The Conclusion section has been corrected
Reviewer 2 Report
Dear Authors,
You have written an interesting study. However, several parts need to be addressed for greater clarity and repeatability. Additionally, the paper needs English proof for better clarity as some sentences are hard to read.
Title: The title does not reflect your study aims. Please rephrase
Abstract: The conclusion in the abstract is poorly written and does not summarise your results. Correct
Line 34 - 35 / There is no connection between these sentences. You just jump from a K1 style to anthropometrical features. Amend
Line 48 - delete ''plan''
Line 53 - ''it is an important element'' according to who? Reference needed.
Line 80-84 / What does hydration have to do with your study? Again, it is not clear as you just jump to a new topic without any link to previously mentioned pH values. Why a reference from judo? Is there no similar research in kickboxing? Correct
Overall, from this introduction, I don-t see the rationale behind the importance of the pH levels and the temperature for sport or particularly kickboxing. Additionally, your hypothesis in lines 93 and 94 is a bit vague as in your results, you don't report the number of ''blows'' - hits. Therefore you can't answer your hypothesis from the start. Rephrase your aim.
Additionally, the introduction has little to no information on technical and tactical skills and their importance to pH. Please address this issue.
Participants - how many from each category? -add
Why such a broad weight category sample? Elaborate
What was their training experience? Report (training and competing is not the same - someone can train for 5 years, however only compete for 3 years). Be specific about your inclusion criteria. Shorten the sample description (muscle mac, BMI, etc). This can be described in 2 lines.
What is the point of reporting daily calorie intake and metabolic age for your study? delete
Line 119 - (WMA, 2013) what is this? Is it a citation? If yes, it is in the wrong format and it is missing from the references.
Sparring bout analysis:
- with how many cameras was this recorded?
-line 147 - you state ''medalist'' this is the first time you mention that? was that the inclusion criteria? elaborate
-line 147 - ''they had won'' so why didn't you analyse the fights they have lost? elaborate
Who carried out the analysis (how many persons)? report
How was skin temperature measured during the rounds when the athletes were sweaty? Did you clean the area and also where on the body did you measure the temperature? Report
Was the device cleaned between different pH measurements? report
How many measurements per area did you take for the reliability of measurements? report
Report the model of the belt for heart rate measurements and it is a chest belt - correct
Line 169-170 / so if the strap had to be put on after the round you didn't measure the HR directly after the bout! Correct the description and be more accurate. How long did it take to put it on? Report
How many people and who participated in these measurements (temperature, HR, pH)? report
In Line 174 you finally describe who was included in the analysis - this needs to be moved in the sample description.
''Fasting body composition'' what did you mean by that? Elaborate
At what time did you perform the body composition analysis (before the official weight in or after that)? report
Table 4 and 5 descriptive statistic is too large - report only mean, SD and 95% CI. Also, you could merge those two tables in 1. Correct
Table 6 - look at the previous comment
Line 245 - 247 / the description of results is inaccurate. Correct
Results description in lines 258-262 are based on which results - which table?
Where is the limitations paragraph? ADD!
Conclusions and practical recommendations are poorly written - rewrite
Overall you are mixing PH values and technical tactical analysis. It looks like you are forcing those two data in one paper and altogether it does not make any sense. I would understand it if you would try to see any connection between ph and technical-tactical analysis, however, you don't do that and your paper is mostly centred on body composition in connection to technical-tactical analysis. Overall the paper does not follow a clear direction as you are jumping from one topic to another. Additionally, the title of the paper is not even close to the content of the paper where you have again Techincal-tactical analysis per weight categories and body composition.
Altogether, I have to recommend the decision of '' Reconsider after major revision'' as the paper does not have a clear introduction and rationale behind the main aim and the methods are not accurately described. However, the data could be of value if they are presented in a clear way by the authors.
Kind regards
Author Response
Dear Reviewer,
Thank you very much for your time and valuable comments, which all have been considered and incorporated. The detailed list of responses is given below. We hope that the modifications and explanation will be acceptable for you.
Yours sincerely,
Rydzik, corresponding author
Title: The title does not reflect your study aims. Please rephrase
A: The title has been rephrased
Abstract: The conclusion in the abstract is poorly written and does not summarise your results. Correct
A: This part has been corrected
Line 34 - 35 / There is no connection between these sentences. You just jump from a K1 style to anthropometrical features. Amend
A: This sentence has been removed because K1 was additionally discussed in previous lines at the request of Reviewer 1
Line 48 - delete ''plan''
A: This part has been corrected
Line 53 - ''it is an important element'' according to who? Reference needed.
A: The sentence has been rearranged. This was the thought of the authors.
Line 80-84 / What does hydration have to do with your study? Again, it is not clear as you just jump to a new topic without any link to previously mentioned pH values. Why a reference from judo? Is there no similar research in kickboxing? Correct
A: This has been corrected (it concerned a literature review)
Overall, from this introduction, I don-t see the rationale behind the importance of the pH levels and the temperature for sport or particularly kickboxing. Additionally, your hypothesis in lines 93 and 94 is a bit vague as in your results, you don't report the number of ''blows'' - hits. Therefore you can't answer your hypothesis from the start. Rephrase your aim.
A: Thanks for your suggestions. It helped us improve our introduction. We made changes to make the paper more readable. We have improved the hypothesis and added an analysis of the average pH level in relation to sports skill level.
Additionally, the introduction has little to no information on technical and tactical skills and their importance to pH. Please address this issue.
A: The description has been extended
Participants - how many from each category? –add
A: Table 1 has been added
Why such a broad weight category sample? Elaborate
A: We wanted to present all weight categories that are present in kickboxing. We attempted to conduct our research as broadly as possible and surveyed the categories we had access to. Other information has been added in the limitation section of the study.
What was their training experience? Report (training and competing is not the same - someone can train for 5 years, however only compete for 3 years). Be specific about your inclusion criteria. Shorten the sample description (muscle mac, BMI, etc). This can be described in 2 lines.
A: More information has been added. We cannot shorten the description as other Reviewers suggested making it longer.
What is the point of reporting daily calorie intake and metabolic age for your study? Delete
A: This has been removed as suggested.
Line 119 - (WMA, 2013) what is this? Is it a citation? If yes, it is in the wrong format and it is missing from the references.
A: WMA stands for World Medical Association, the organization that developed the Declaration of Helsinki - the standard methodological abbreviation
Sparring bout analysis:
- with how many cameras was this recorded?
A: More information has been added
-line 147 - you state ''medalist'' this is the first time you mention that? was that the inclusion criteria? Elaborate
A: Thank you for your comment, this was a mistake that has been corrected
-line 147 - ''they had won'' so why didn't you analyse the fights they have lost? Elaborate
A: Thank you for your comment, this was a mistake that has been corrected
Who carried out the analysis (how many persons)? Report
A: Information has been added
How was skin temperature measured during the rounds when the athletes were sweaty? Did you clean the area and also where on the body did you measure the temperature? Report
A: The area was cleaned, we have supplemented this part.
Was the device cleaned between different pH measurements? Report
A: After each measurement, the probe was immersed in distilled water.
How many measurements per area did you take for the reliability of measurements? Report
A: One measurement was taken in each measurement area according to the testing procedure
Report the model of the belt for heart rate measurements and it is a chest belt – correct
A: Information has been added
Line 169-170 / so if the strap had to be put on after the round you didn't measure the HR directly after the bout! Correct the description and be more accurate. How long did it take to put it on? Report
A: This part has been corrected and we have added information in the Limitations section.
How many people and who participated in these measurements (temperature, HR, pH)? report
A: The measurements were made by the authors of the study (three people)
In Line 174 you finally describe who was included in the analysis - this needs to be moved in the sample description.
A: Thank you for your comment, this was a mistake that has been removed
''Fasting body composition'' what did you mean by that? Elaborate
A: This part has been corrected
At what time did you perform the body composition analysis (before the official weight in or after that)? Report
A: The information has been added
Table 4 and 5 descriptive statistic is too large - report only mean, SD and 95% CI. Also, you could merge those two tables in 1. Correct
A: We would suggest maintaining these values for a more complete picture of the measurements, but if you feel it is necessary we will remove them
Table 6 - look at the previous comment
A: We would suggest maintaining these values for a more complete picture of the measurements, but if you feel it is necessary we will remove them
Line 245 - 247 / the description of results is inaccurate. Correct
A: This part has been corrected
Results description in lines 258-262 are based on which results - which table?
A: The results do not refer to any table, they are presented in the text due to a large number of tables in the manuscript. The table is attached in the supplementary material
Where is the limitations paragraph? ADD!
A: The paragraph has been added.
Conclusions and practical recommendations are poorly written – rewrite
A: rewritten conclusions
Overall you are mixing PH values and technical tactical analysis. It looks like you are forcing those two data in one paper and altogether it does not make any sense. I would understand it if you would try to see any connection between ph and technical-tactical analysis, however, you don't do that and your paper is mostly centred on body composition in connection to technical-tactical analysis. Overall the paper does not follow a clear direction as you are jumping from one topic to another. Additionally, the title of the paper is not even close to the content of the paper where you have again Techincal-tactical analysis per weight categories and body composition.
A: Thank you, your comments have significantly improved the quality of our manuscript. We would like to provide a comprehensive analysis of body composition and epidermal changes in the same athletes. A paper on body composition alone would not have introduced scientific novelty, so we decided to present it in this form.
Altogether, I have to recommend the decision of '' Reconsider after major revision'' as the paper does not have a clear introduction and rationale behind the main aim and the methods are not accurately described. However, the data could be of value if they are presented in a clear way by the authors.
A: Thank you for your valuable comments . Manuscript has been proofread by a native speaker
Round 2
Reviewer 2 Report
Dear Authors,
Thank you for addressing most of the raised questions and issues. However, some small parts still need to be addressed.
Table 1 caption is written in Polish. Translate to English and delete the last % colum.
In Table 3 you still have some Polish words. Correct
Overall, in my opinion, the authors have substantially improved their paper. Therefore, I recommend acceptance after minor revision.
Kind regards